

# LIU-NET: lightweight Inception U-Net for efficient brain tumor segmentation from multimodal 3D MRI images

Gul e Sehar Shahid[1], Jameel Ahmad[2], Chaudary Atif Raza Warraich[3], Amel Ksibi[4], Shrooq Alsenan[4], Arfan Arshad[2], Rehan Raza[5] and Zaffar Ahmed Shaikh[6,7]

[1] Department of Artificial Intelligence, University of Management & Technology, Lahore, Pakistan
[2] Department of Computer Science, School of Systems and Technology, University of Management & Technology, Lahore, Pakistan
[3] Department of Computer Science, COMSATS Institute of Information Technology, Lahore, Pakistan
[4] Department of Information Systems, College of Computer and Information Sciences, Princess Nourah bint Abdulrahman University, Riyadh, Saudi Arabia
[5] School of Information Technology, Murdoch University, Perth, Australia
[6] Department of Computer Science and Information Technology, Benazir Bhutto Shaheed University Lyari, Karachi, Pakistan
[7] School of Engineering, École Polytechnique Federale de Lausanne, Lausanne, Switzerland

Corresponding author
Zaffar Ahmed Shaikh,
zashaikh@bbsul.edu.pk

## ABSTRACT

Segmenting brain tumors is a critical task in medical imaging that relies on advanced deep-learning methods. However, effectively handling complex tumor regions requires more comprehensive and advanced strategies to overcome challenges such as computational complexity, the gradient vanishing problem, and variations in size and visual impact. To overcome these challenges, this research presents a novel and computationally efficient method termed lightweight Inception U-Net (LIU-Net) for the accurate brain tumor segmentation task. LIU-Net balances model complexity and computational load to provide consistent performance and uses Inception blocks to capture features at different scales, which makes it relatively lightweight. Its capability to efficiently and precisely segment brain tumors, especially in challenging-to-detect regions, distinguishes it from existing models. This Inception-style convolutional block assists the model in capturing multiscale features while preserving spatial information. Moreover, the proposed model utilizes a combination of Dice loss and Focal loss to handle the class imbalance issue. The proposed LIU-Net model was evaluated on the benchmark BraTS 2021 dataset, where it generates remarkable outcomes with a Dice score of 0.8121 for the enhancing tumor (ET) region, 0.8856 for the whole tumor (WT) region, and 0.8444 for the tumor core (TC) region on the test set. To evaluate the robustness of the proposed architecture, LIU-Net was cross-validated on an external cohort BraTS 2020 dataset. The proposed method obtained a Dice score of 0.8646 for the ET region, 0.9027 for the WT region, and 0.9092 for the TC region on the external cohort BraTS 2020 dataset. These results highlight the effectiveness of integrating the Inception blocks into the U-Net architecture, making it a promising candidate for medical image segmentation.

**How to cite this article** Shahid GeS, Ahmad J, Warraich CAR, Ksibi A, Alsenan S, Arshad A, Raza R, Shaikh ZA. 2025. LIU-NET: lightweight Inception U-Net for efficient brain tumor segmentation from multimodal 3D MRI images. *PeerJ Comput. Sci.* 11:e2787 **DOI** 10.7717/peerj-cs.2787

## INTRODUCTION

Brain tumors (BTs) constitute a significant factor in the global mortality rate, and according to Cure Brain Cancer Foundation data, among all types of malignancies worldwide, brain tumors cause more fatalities in individuals under the age of 40 than any other type of cancer (*Elmezain et al., 2022*; *Ali et al., 2022*). This statistic underscores the pressing need for continued research, awareness, and robust support to mitigate the devastating effects of brain tumors on young generations. BTs consist of a bunch of irregular cells within the human brain. These abnormal cells have the potential to affect the nervous system adversely. This effect causes harm to the surrounding healthy brain tissues. The brain is crucial in facilitating communication and coordination among different body components (*Tarasiewicz, Kawulok & Nalepa, 2021*; *Akram et al., 2025*). Tumors disrupt the function of the brain and are the most critical conditions afflicting the human system (*Baid et al., 2021*). In 2020, it was estimated that 251,329 individuals worldwide tragically lost their lives because of cancerous tumors in the brain. BTs are broadly classified into two main categories: primary and secondary brain tumors. Primary BTs consist of cells that originate within the brain. On the other hand, metastatic BTs, often called secondary BTs, comprise cells that originate in other parts of the body and migrate to the brain through the bloodstream (*Liu et al., 2023*). BTs are further classified based on the nature of tumorous cells, with distinctions between malignant and non-malignant forms depending on their severity. One of the most common types of primary brain tumors, called gliomas, begins its growth within the brain. According to the "World Health Organization (WHO)" report, this primary tumor is categorized into grades one to four. This grading system is determined by the tumor's behavior and microscopic characteristics, with grade one being the least aggressive and grade four representing the most aggressive type of tumor (*Yousef et al., 2023*). Treating gliomas typically involves a combination of therapeutic approaches, including chemotherapy, radiation therapy, and surgical interventions. However, detecting and segmenting gliomas poses a significant challenge for radiologists due to the variability in tumor size and their diverse locations within the brain. The effective treatment of a tumor heavily relies on factors such as its type, location, and grade. Consequently, tumor segmentation plays a significant role in the treatment planning process (*Zhang et al., 2023*). LIU-Net solves these issues by making computers more efficient and speedier with a lightweight architecture. With multiscale feature extraction, it picks up on several tumor characteristics. Specialized loss algorithms fix the class imbalance. This improves brain tumor segmentation in complex datasets (*Zhang et al., 2020b*).

### Contribution

The main contributions of the study are summarized as follows:

- LIU-Net, a lightweight deep learning-based encoder-decoder architecture, can handle longer training times and model parameters. This architecture combines the skip connections of U-Net with the strengths of Inception to facilitate the extraction of 3D magnetic resonance imaging (MRI) multi-scale features.

- The LIU-Net balances computational economy and segmentation accuracy, making it suited for resource-constrained clinical and research environments.

## Motivation and innovation

Medical imaging is indispensable in diagnosing and managing tumors, including the critical task of tumor segmentation and assessing the effectiveness of therapy. Common medical imaging modalities encompass ultrasounds, computed tomography (CT), MRI scans, and X-ray image examinations. Among these, MRI is the most widely utilized modality in brain tumor diagnosis. Its acclaim stems from its exceptional sharpness, tissue resolution, and versatility in adjusting various parameters to obtain specific anatomical details (*Bukhari & Mohy-ud Din, 2021*; *Akhund et al., 2024*). MRI is an advanced imaging technique that relies on the principles of nuclear magnetic resonance (*Sariturk & Seker, 2022*). MRI utilizes powerful magnetic fields, typically ranging from 1.5 Tesla (T) to 3 Tesla (T), in conjunction with radio frequency waves to generate highly detailed images of internal body structures and tissues (*Pei & Liu, 2021*). MRI is a non-invasive imaging technique employed to gather data about brain cells without posing any harm to the organ under examination. This method is entirely devoid of high ionization or radiation effects. MRI is preferred for brain imaging because it offers high-resolution images without the harmful effects of ionizing radiation (*Mehak, Muneer & Nawab, 2023*; *Yang et al., 2024*). In contrast, ultrasound, X-rays, and CT scans have their applications in medical imaging. However, they are not the preferred choice for brain examination due to limitations in image quality, safety concerns, and the specific needs of brain imaging. MRI encompasses various types of image modalities or sequences, with T1-weighted (T1w), T1-weighted contrast-enhanced (T1ce), T2-weighted (T2w), and fluid-attenuated inversion recovery (FLAIR) being among the most extensively employed modalities for diagnosing brain tumors. T1w imaging primarily serves to assess healthy tissues. Conversely, T2w imaging accentuates the bright tumor area, whereas T1ce imaging underscores the bright tumor boundary. The FLAIR scan is crucial in distinguishing between edema and cerebrospinal fluid (*Gad, Soliman & Darweesh, 2023*). The LIU-Net was chosen because it could deal with important problems in brain tumor segmentation, like class mismatch and computational complexity. LIU-Net is different from other models because it uses both Inception-style convolutional blocks and the U-Net framework. This enables it to record multiscale features quickly while keeping the computational costs low. On-time and accurate brain tumor segmentation (BTS) using the described four modalities is instrumental in empowering medical professionals to conduct tumor surgeries with safety and precision. This ensures that healthy brain regions remain unharmed during the surgical procedure. Automated segmentation of brain tumors from MRI images has the potential to significantly accelerate radiologists' workflow and improve result consistency (*Liu et al., 2023*). Nonetheless, automating the segmentation of brain tumors and their sub-regions presents a formidable challenge, primarily due to the unpredictable nature of tumorous cells. These cells can manifest in diverse locations within brain tissues, exhibiting variations in size, appearance, and shape (*Baid et al., 2020*). One of the notable techniques

in computer vision is the utilization of convolutional neural networks (CNN), which demonstrate the ability to capture high-dimensional hierarchical features independently. Conversely, classical machine learning algorithms depend on manually extracted feature engineering, distinct from the automated feature learning capabilities inherent in CNNs. Inception-style blocks in the U-Net design allow LIU-Net to swiftly acquire multiscale characteristics from tumors of various sizes and forms. Because it balances model complexity and processing load, LIU-Net can accurately divide brain tumors in real-time even with limited resources. This study presents an innovative deep-learning model for segmenting brain tumors called LIU-Net that combines a lightweight Inception network within the U-Net architecture, offering a unique and effective approach to this task (*Khan et al., 2022*; *Laghari et al., 2024*). This combination speeds up training and ensures robustness in a lightweight framework. The lightweight Inception U-Net is designed to provide good accuracy while using fewer computer resources. This makes it helpful for medical imaging, particularly in situations where resources are limited.

## Structure of paper

The structure of this research article is as follows: the literature review, delving into the existing body of knowledge, is discussed in "Literature Review". Moving forward, "Materials and Methods" presents the proposed LIU-Net model, which is the central focus of this research study. Different experiments are conducted in "Results and Analysis", and a detailed discussion of the obtained results is provided. Finally, "Conclusion and Future Directions" serves as the conclusion, wrapping up this article with potential future directions.

## LITERATURE REVIEW

Brain tumors represent a serious medical condition that requires early detection and swift intervention to enhance the prospects of a favorable outcome (*Ranjbarzadeh et al., 2024*; *Kousar et al., 2024*). Before deep learning techniques, traditional machine learning methods were employed to acquire insights from brain images, relying on manually crafted feature engineering. Recently, deep learning has emerged as a notable advancement in medical imaging. Deep learning models independently capture both local and global information in medical imaging. This section offers an in-depth literature review of contemporary approaches rooted in architectural frameworks for brain tumor segmentation.

## U-Net based segmentation

Manual segmentation is a time-consuming and intensive process that heavily relies on the expertise and experience of radiologists. Experts and medical professionals now demand automated segmentation from volumetric brain scans. Numerous solutions have been put forward to address the automated brain tumor segmentation from 3D MRI images. The "U-Net model" is widely recognized as a prominent deep learning model uniquely designed for biomedical image segmentation. The U-Net architecture was originally

introduced by *Ronneberger, Fischer & Brox (2015)* and gained widespread recognition for its capacity to produce reliable segmentation outcomes, particularly in scenarios with limited training data. The U-Net architecture is characterized by its distinctive U-shaped structure, consisting of a left contracting portion known as the encoder, which is responsible for feature extraction. On the other hand, it incorporates a right-expanding segment called the decoder, which utilizes these features to generate an output mapped back to the original image pixels. U-Net utilizes concatenation to relay important contextual information and feature maps from the encoder to the decoder, enabling the classifier to make well-informed predictions. In the domain of brain tumor segmentation, most existing techniques have employed either 2D or 3D convolutions in the training of deep CNN models (*Zheng et al., 2024*; *Liu et al., 2024*).

*Dong et al. (2022)* employed a model called a 2D U-Net for segmenting individual slices within each 3D mpMRI volume. This approach was characterized by its speed in both phases *i.e.*, training and testing, along with lower computational demands. The evaluation of this approach was conducted using the BraTS 2015 dataset. This dataset consists of a total of 220 cases of high-grade brain tumors and 54 cases of low-grade tumors. Nevertheless, it's important to highlight that this approach was notably over-parameterized. This approach boasts around 35 million parameters. It also did not efficiently exploit the 3D contextual information present in the dataset. *Isensee et al. (2021)* employed an ensemble approach involving 3D U-Nets that were trained on an extensive dataset. The architecture was trained and assessed using the training datasets from both "BraTS 2017 and BraTS 2015". Their efforts primarily concentrated on implementing minor refinements to achieve competitive segmentation performance. *Wang et al. (2021)* introduced TransUNet as an innovative approach to leverage transformers in the area of medical image segmentation. The TransUNet architecture bears a resemblance to the well-established U-Net (*Ranjbarzadeh et al., 2024*) framework, wherein CNNs (convnets) serve as feature extractors, while transformers play a pivotal role in encoding global context. However, a key characteristic shared by TransUNet and its subsequent iterations (*Ali, Kako & Abdi, 2022*) is the treatment of convnets as the primary backbone, upon which transformers were applied to capture extended contextual relationships. While effective, this approach presents a potential challenge: it may not fully harness the advantages of transformers. This research uses a synapse multi-organ segmentation dataset. At its core, this constraint arises from the recognition that utilizing only a limited number of transformer layers may prove insufficient in effectively integrating extended contextual relationships with the convolutional features inherent in the dataset. Convolutional representations often provide hierarchical concepts and exact spatial information. Thus, transformers and convnets can be synergised to better segment medical images.

## Lightweight models in medical image segmentation

Traditional U-Net, 3D U-Net, and TransUNet are well-established in medical image segmentation. More recently, lightweight deep learning models that balance accuracy and computing efficiency have been developed. These models excel in real-time medical

applications that require quick choices with limited resources. These models include VT-Unet. It simplifies and is accurate in new ways. Self-attention layers and Fourier position coding allow VT-Unet to extract local and global information from 3D medical images with fewer parameters. The ability to quickly process vast amounts of data makes it ideal for brain tumor segmentation. In its skip connections, Lightweight U-Net uses Vision Transformer layers to focus on global feature models while training efficiently. This model is ideal for accurate segmentation jobs like brain tumor detection. BraTS 2021 and other smaller datasets work nicely with it. These lightweight models outperform 3D U-Net and TransUNet designs. This is especially true for fewer parameters and faster training. TransUNet works successfully because it uses transformers and CNNs. But devices like VT-Unet segment better with less computer power, making them suitable for clinical settings with restricted computational capacity. This study uses these recent brain tumor segmentation advances to maximize lightweight constructions' accuracy and practicality (*Cahall et al., 2021*).

Many existing techniques in brain tumor segmentation have focused on improving their performance. While current segmentation models have shown promising results, they often struggle to fully utilize the potential of 3D volumetric brain tumor scans due to limitations in computational resources. Numerous studies within the literature review section have emphasized notable enhancements in segmentation outcomes through alterations to the model's architecture. One key issue observed is the problem of over-parameterization, which leads to complex models and longer training times. This research introduces a novel, lightweight deep learning model with minimal parameters to address this challenge. This research aims to increase the accuracy and authenticity of brain tumor segmentation tasks while mitigating issues related to prolonged training periods and the use of numerous hyperparameters (*Khan et al., 2023*). For the solution, this research study introduces an innovative approach that combines the Inception network with a 3D U-Net model, strategically addressing concerns about training efficiency and effectiveness in brain tumor segmentation. This integrated approach is expected to streamline and improve the segmentation process for brain tumors. Table 1 describes the proposed methodology, novelty, and results of the literature described below.

## Comparison with prior works

Compared to other works that combine U-Net with Inception (*e.g.*, *Cahall et al. (2021)*, *Sariturk & Seker (2022)*, *Yang et al. (2024)*, *Zhang et al. (2020b)*), our proposed LIU-Net sets itself apart from models that blend U-Net with Inception by prioritizing efficiency without compromising segmentation accuracy. Like previous approaches that enhance Inception layers with more complex parameters; instead, LIU-Net incorporates efficient multi-kernel parallel convolutions to lower computational expenses while upholding segmentation performance. In addition, by combining the Dice and Focal loss techniques, we learned that this approach has not been extensively explored in previous research, which leads to higher-level handling of class imbalance. Lastly, the validity of the LIU-Net is checked on both the BraTS 2021 and BraTS 2020 datasets, and it show the potential to generalize its performance across different datasets.

**Table 1 Comparison of existing brain tumor segmentation methods.**

| Study | Proposed methodology | Novelty | Results |
|---|---|---|---|
| Dong et al. (2022) | 2D U-Net for 3D mpMRI slices | Fast training/testing; lower computational demand | Effective but over-parameterized (35 million parameters); lacks 3D context utilization |
| Isensee et al. (2021) | Ensemble of 3D U-Nets | Minor refinements on BraTS datasets | Improved accuracy, but computationally intensive |
| Chen et al. (2023) | TransUNet with CNNs and transformers | Integrates transformer layers for global context | Effective but limited by a small number of transformer layers; spatial information not fully exploited |
| Karimi, Vasylechko & Gholipour (2021) | Convolution-free segmentation model using transformers | Flattened image input; hierarchical object concept acquisition | Handles multiscale data but requires pre-training on large datasets |
| Ma et al. (2023) | Hierarchical transformer blocks in U-Net architecture | Encoder-decoder transformers; multiscale feature learning | Significant accuracy improvements, but complex and computationally demanding |
| Jia & Shu (2022) | Vision Transformer layers in skip connections | Global feature modeling through deep connections | Accurate but parameter-heavy; evaluated on BraTS 2021 |
| Peiris et al. (2022) | Lightweight 3D U-Net with self-attention | Window-based self-attention; Fourier position coding | High accuracy with fewer parameters; suitable for real-time use |
| Proposed method (LIU-Net) | Lightweight Inception-based U-Net for brain tumor segmentation | Combines Inception-style blocks with U-Net for efficient feature extraction | Superior segmentation accuracy on BraTS 2021 dataset; efficient and practical for real-time applications |

# MATERIALS AND METHODS

This article introduces lightweight Inception U-Net (LIU-Net), a new deep learning brain tumor segmentation model. This model uses the Inception network's multi-scale trait capture and the U-Net architecture's segmentation. We combine these two networks to attain excellent segmentation accuracy with little computational power. LIU-Net efficiently separates brain tumors from 3D MRIs. It avoids standard deep learning model issues like too many parameters and slow training. Statistically, LIU-Net outperforms U-Net in segmenting the BraTS 2021 dataset using paired t-tests. Within this section, this study provides comprehensive information concerning the dataset, elucidates the pre-processing steps executed on it, and delineates the specific details of the implementation process for the proposed methodology. This section also delves extensively into the particulars of the proposed LIU-Net model, offering a comprehensive explanation of the employed loss functions throughout the training process. Additionally, Fig. 1 illustrates the general workflow of the proposed system model.

## Dataset

The LIU-Net model was implemented utilizing the openly accessible BraTS 2021 challenge dataset. External cohort validation was carried out to validate the effectiveness of the suggested model, and an assessment was performed using the BraTS 2020 dataset. The datasets employed in this research were obtained from the Medical Image Computing and Computer-Assisted Intervention (MICCAI) Multimodal Brain Tumor Segmentation Challenge (BraTS). These datasets were meticulously curated by healthcare professionals

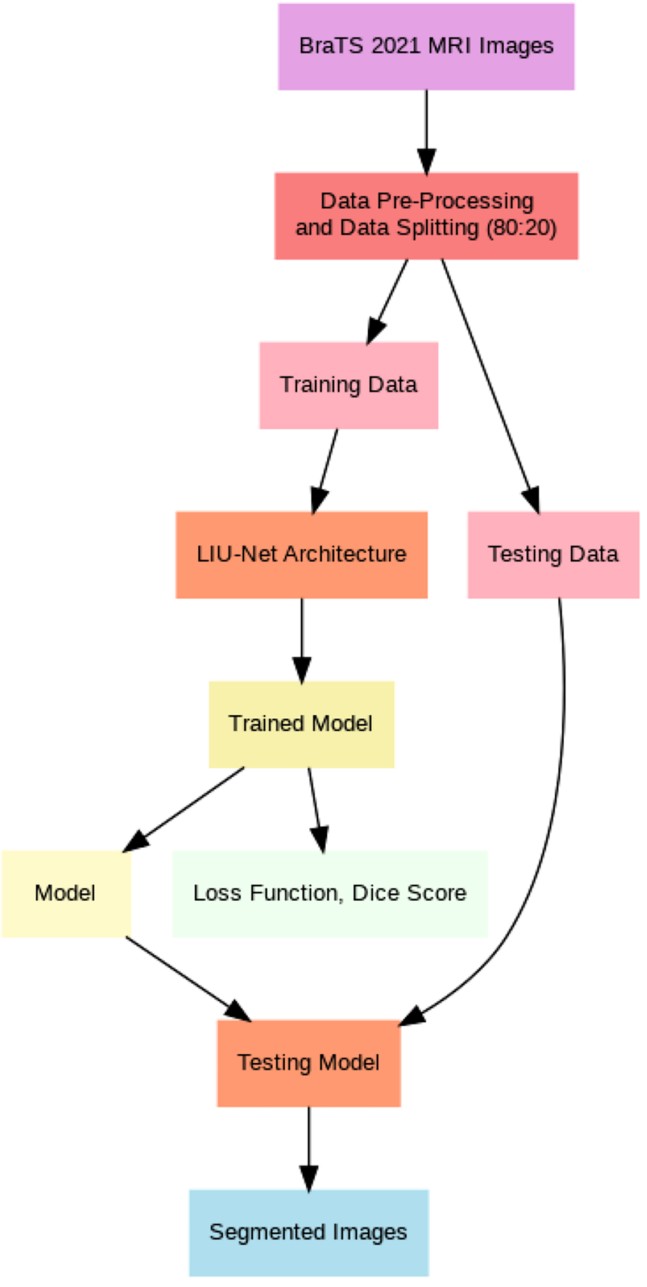

**Figure 1 General flow of proposed architecture.**

affiliated with the University of Pennsylvania and UPenn's Center for Biomedical Image Computing and Analysis (CBICA) (*Baid et al., 2021*).

The BraTS 2021 training dataset comprises 3D MRI brain scans collected from 1,250 patients. Each image within the dataset has dimensions of $240 \times 240$ pixels, and a complete 3D scan consists of 155 slices. The patient dataset encompasses four unique MRI modalities: FLAIR, T1, T2, and T1ce. Figure 2 depicts example subjects drawn from the BraTS 2021 dataset. Significantly, these modalities primarily diverge with respect to the brain's water content, particularly cerebrospinal fluid, along with variances in tissue

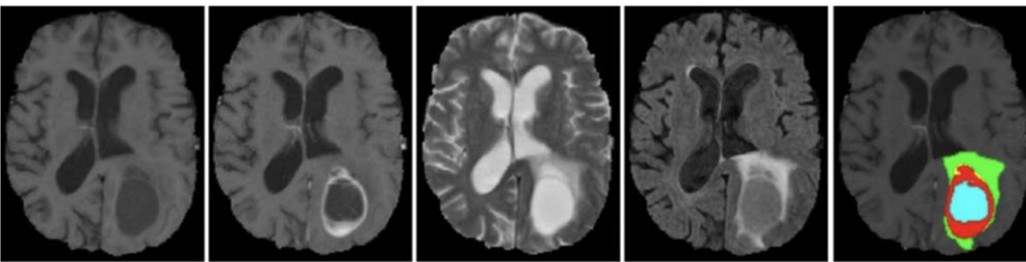

**Figure 2 Sample MRI images from BraTS 2021 dataset.**

**Table 2 Dataset split of BraTS 2021.**

| Dataset | Training set | Validation set | Testing set |
|---|---|---|---|
| BraTS 2021 | 70% | 10% | 20% |
| Total patients = 1,250 | 875 | 125 | 250 |

intensities. The segmentation mask is meticulously crafted through manual annotation by a group of neuroradiologists, ranging from one to four experts (*Raza et al., 2023*). This dataset encompasses four primary classes: background (labeled as 0), necrosis and non-enhancing tumor (labeled as 1), edema (labeled as 2), and ET (initially labeled as 4). However, for the implementation, label 4 was transformed into label 3. Before the training of the proposed model, the dataset was class balance split into training, validation, and testing subsets. Specifically, 70% of the data is allocated to the training set, 10% to the validation set, and the remaining 20% to the testing set, as outlined in Table 2.

The BraTS 2020 dataset consists of 3D MRI brain scans acquired from a cohort of 369 individuals diagnosed with glioma. Among these cases, 76 individuals were identified as having low-grade glioma (LGG), while the remaining scans were sourced from high-grade glioma (HGG) patients. Every MRI scan in this dataset has a consistent dimension of 240 × 240 pixels and 155 slices. Each patient's dataset comprises four distinct MRI modalities: T1, T1ce, T2, and FLAIR. This comprehensive multimodal approach provides a detailed perspective on the glioma cases in this dataset for external cohort validation. The sample images from the BraTS 2020 dataset are shown in Fig. 3.

## Pre-processing

The competition organizers have completed various preprocessing steps on both the BraTS 2021 and BraTS 2020 datasets prior to their public release to ensure their integrity and originality. The images have undergone prior processing, including co-registration, skull-stripping, alignment into a standardized space, and achieving isotropic tenacity (*Zhang et al., 2020a*). MRI intensity varies due to scanner magnetic fields. To improve segmentation results, data must be pre-processed before model training (*Shaikh, 2018*). This study preprocessed MRI images by normalizing their pixel values to improve comparability and analysis. Normalization is a common data preprocessing method in

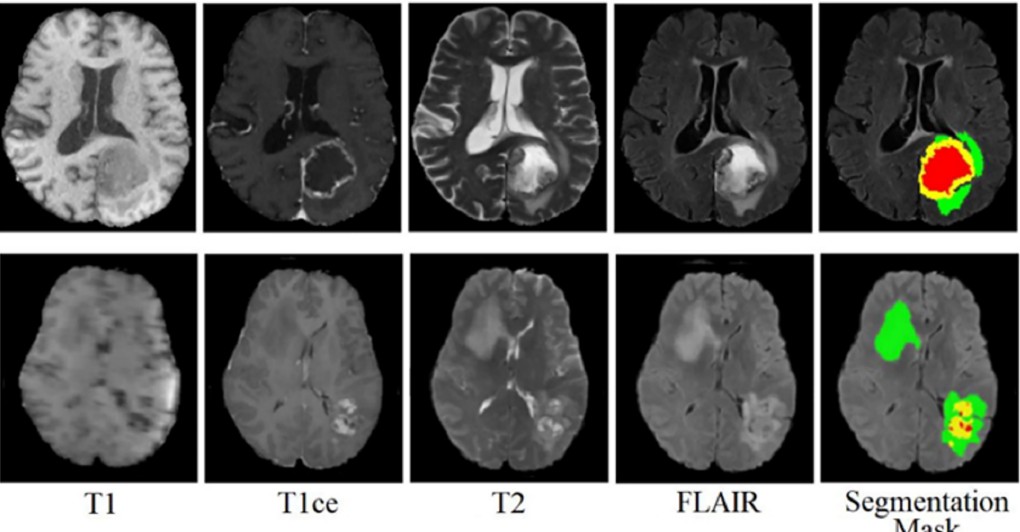

**Figure 3 Sample MRI images from the BraTS 2020 dataset.**

medical image analysis that standardizes pixel values across modalities. The normalization process used the **scaler.fit_transform** method, which independently transforms each MRI modality's pixel values. This normalization step is crucial as it reduces the impact of variations in pixel intensity between different scans and enhances the robustness and convergence of subsequent image processing and machine learning algorithms. It also contributes to improved interpretability and generalization of the results. The zero mean and unit variance normalization, also called Gaussian distribution, is applied to all images, and it can be computed by Eq. (1).

$$X_{\text{norm}} = \frac{X - X_{\min}}{X_{\max} - X_{\min}} \tag{1}$$

where $X$ represents the original data, $X_{\text{norm}}$ represents the normalized data, $X_{\min}$ represents the minimum value in the original data and $X_{\max}$ represents the maximum value in the original data.

A resizing operation has been conducted on all MRI images, transitioning them from their initial $240 \times 240 \times 155$ dimensions to a more manageable size of $128 \times 128 \times 128$ because of the limited computational power. To fully harness the valuable information embedded within the four modalities *i.e.*, T1, T1ce, T2, and FLAIR, they have been integrated through stacking, allowing us to capitalize on the strengths of each modality. Throughout the training phase, input instances measuring $128 \times 128 \times 128 \times 4$ (with "4" denoting the four modalities) have been employed to train the model (*Shaikh & Khoja, 2014*). This strategy guarantees that the model is exposed to the entirety of the available data spectrum during the training process. Even though the method makes the images smaller, it does not hurt the diagnostic process. The model still gets useful information from all four modes (T1, T1ce, T2, and FLAIR) even though they are stacked into a single input channel. This integration enables the model use traits from all four images that work

well together while still keeping the resolution high enough for effective segmentation. Resizing also makes sure that the model can easily process multiple 3D images, which speeds up training without losing important diagnostic details. There are several methods built into the model's design, such as Inception blocks and loss functions, that help keep the segmentation accuracy high even after the model is resized. In this way, resizing makes computations faster without lowering the quality of the diagnostic results.

## Proposed LIU-Net architecture

The basic U-Net model can be made better by adding LIU-Net blocks that look like Inception and have parallel convolutional pathways with variable kernel sizes. This design lets the network detect little details and huge spatial patterns. This is notably useful for brain tumor fragmentation. Unlike earlier methods that use deeper networks to improve performance, our approach increases representational capacity without raising parameters. This maximizes training efficiency and reduces overfitting. The U-Net architecture is renowned and widely adopted within the field of biomedical research, serving as a prominent and extensively employed framework for semantic image segmentation (*Khan et al., 2021*; *Shaikh et al., 2022*). The U-Net architecture has received acclaim for its effective utilization of local and global feature extraction techniques across various scales. Furthermore, using skip connections, the U-Net design seamlessly integrates a mechanism through which feature maps from encoder levels are transmitted to their corresponding decoder levels. When generating the segmentation mask, this design empowers the segmentation classifier to adeptly consider fine-grained details associated with edges and boundaries and higher-level contextual information about tumorous characteristics and shapes. But alongside, certain limitations exist that can complicate the training process. Primarily, as one delves deeper into the network architecture, challenges related to prolonged training times and managing an extensive set of hyperparameters become increasingly prominent. This study, taking the motivation from the U-Net model, introduces LIU-Net architecture for brain tumor segmentation as a solution to address the challenges. The complete architecture of the proposed LIU-Net architecture is represented in Fig. 4. Traditional U-Net architectures excel at segmentation tasks, but they struggle with long learning times and a large number of parameters. Also, their fixed convolutional kernel sizes might make it harder to catch multi-scale features well. Inception-style blocks in the U-Net framework allow for simultaneous feature acquisition at multiple scales. Parallel convolutional layers with kernel sizes from $1 \times 1 \times 1$ to $3 \times 3 \times 3$ to $5 \times 5 \times 5$ should be used. This multi-scale technique helps the network identify small, medium, and big structures, which is crucial for medical image segmentation.

The foundation of this proposed model relies on the U-Net architecture, featuring an encoder-decoder structure with five-level depth. The encoder reduces the resolution of the input MRI images to extract high-level features, whereas the decoder increases the resolution of the feature maps to produce the segmentation mask. The proposed LIU-Net with Inception-style blocks extends this architecture by incorporating additional features for improved feature representation. The key innovation in the LIU-Net model is using Inception-style blocks within each network level. These blocks consist of three parallel

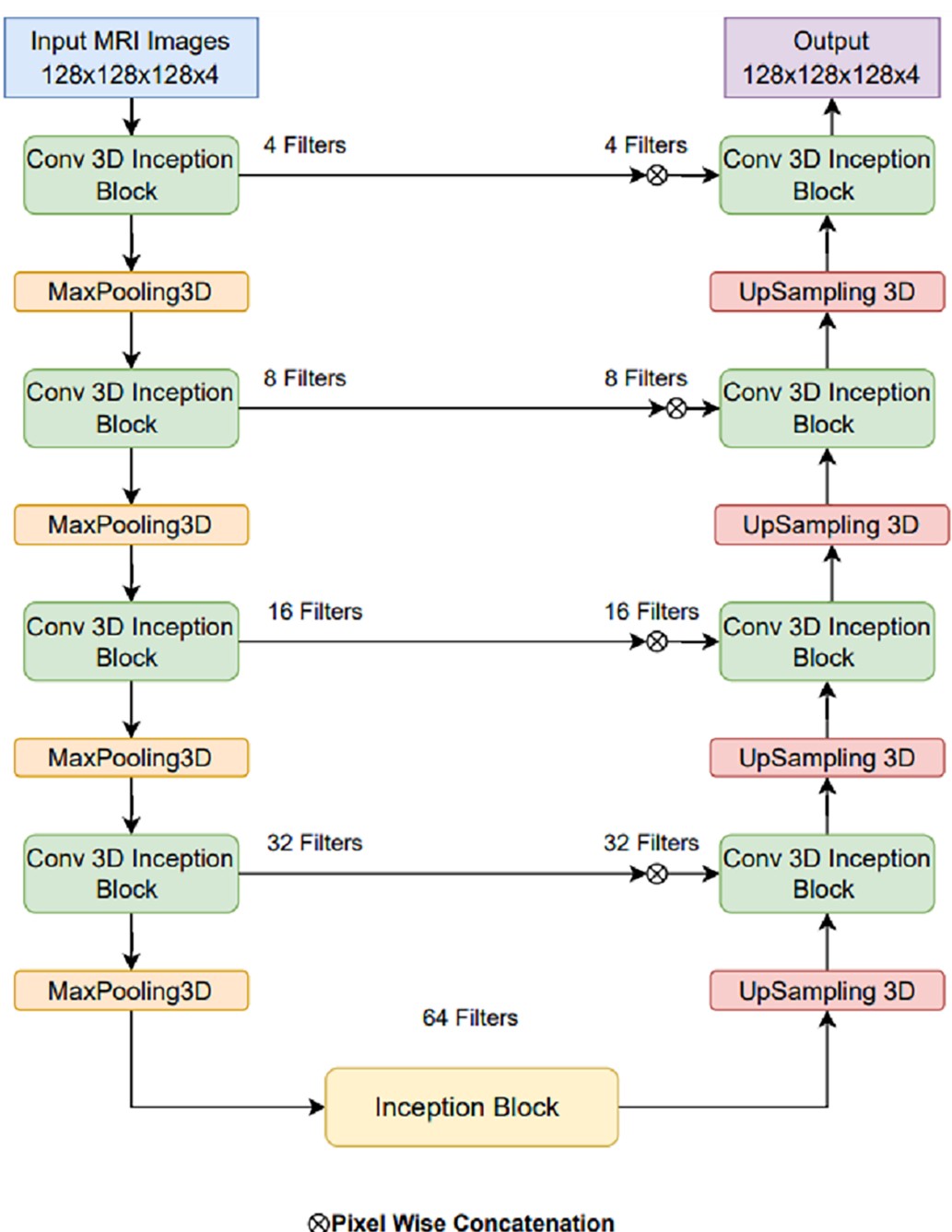

**Figure 4 Proposed LIU-Net architecture.**

convolutional pathways with different kernel sizes. The Inception block used in this study is shown in Fig. 5. Inception-style blocks combine features from different receptive fields to acquire more spatial data than U-Net layers. That makes feature maps stronger at distinguishing things. The parallel Inception blocks of the LIU-Net distribute computing burdens more evenly. This speeds training and reduces over-parameterization compared to deeper networks. LIU-Net is ideal for real-world medical imaging workloads with

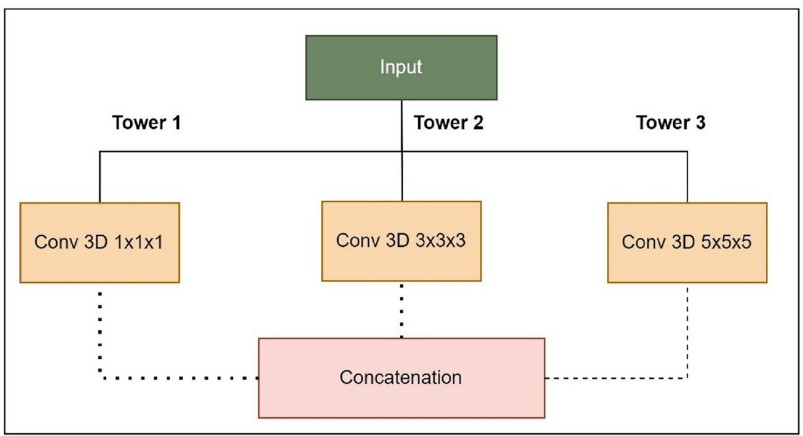

**Figure 5 Expanded Inception blocks.**

limited computer capacity since it reduces computation costs and maximizes feature extraction.

- Tower 1: $1 \times 1 \times 1$ convolutions to capture fine details.
- Tower 2: $3 \times 3 \times 3$ convolutions for capturing medium-sized structures.
- Tower 3: $5 \times 5 \times 5$ convolutions for capturing larger structures.

Combining pathway feature maps helps the network capture more spatial information and learn more discriminative features. This improves the model's ability to handle medical image object sizes and shapes. The proposed LIU-Net has five levels with Inception-style blocks and maxpooling3D downsampling layers. A hierarchical structure makes the network record features at multiple scales, enhancing segmentation accuracy. The network receives a $128 \times 128 \times 128 \times 4$ 3D volumetric image. Every level of feature maps uses the Inception-style block to improve representation. Max-pooling layers emphasise important information by reducing spatial dimensions. The decoder recovers spatial features with up-sampling layers. Skip links between encoder and decoder layers allow fine-grained spatial information to be transmitted during up-sampling. These linkages preserve contextual information for accurate segmentation. This work's design uses Inception-style blocks to elegantly solve training time and parameter complexity issues. This study improved model representation without deeper networks. Interestingly, this architectural choice preserves nuanced aspects while training efficiently. On graphic processing units (GPUs) and tensor processing units (TPUs), Inception-style blocks' parallel convolutional pathways spread computational weight, speeding training. Reducing parameters compared to classic deeper networks reduces over-parameterization concerns. This architectural change enhances segmentation accuracy and keeps the model suitable for medical image analysis. The LIU-Net balances top-tier results with computational restrictions in deep learning for medical imaging. In pseudo-code, the proposed architecture's implementation code is described in Algorithm 1:

| | Algorithm 1   LIU-Net Pseudo-code. |
|---|---|
| 1: | **Step 0:** Read BraTS 2021 training dataset |
| 2: | **Step 1:** Split data into training set (70%), validation set (20%), and testing set (10%) |
| 3: | **Input:** Training dataset $(X_{\text{train}}, Y_{\text{train}})$ |
| 4: | **Input:** Validation dataset $(X_{\text{val}}, Y_{\text{val}})$ |
| 5: | **Input:** Testing dataset $(X_{\text{test}}, Y_{\text{test}})$ |
| 6: | **Input:** Model parameters $\theta$ |
| 7: | **start** |
| 8: | **Step 2:** Preprocessing (Z-score normalization) |
| 9: | Preprocess: Apply Z-score normalization to all datasets $(X_{\text{train}}, X_{\text{val}}, X_{\text{test}})$ |
| 10: | **Step 3:** Train the LIU-Net model |
| 11: | **for** epoch = 1 to $N_{\text{epoch}}$   **do** |
| 12: |    **for** batch = 1 to $N_{\text{train}_{\text{batches}}}$   **do** |
| 13: |       **Step 3.1:** Forward pass and calculate loss (Dice Loss) |
| 14: |       Calculate Loss (Dice Loss) for current batch |
| 15: |    **end for** |
| 16: | **end for** |
| 17: | **Step 4:** Save the trained LIU-Net model parameters |
| 18: | Save: LIU-Net model parameters |
| 19: | **Step 5:** Evaluate on the test set |
| 20: | Evaluate: Test the model on the test set $(X_{\text{test}}, Y_{\text{test}})$ |
| 21: | Calculate: Dice score on the test set |
| 22: | **Step 6:** Segment the test set |
| 23: | **for** each sample $X_t$ in $X_{\text{test}}$   **do** |
| 24: |    $Y_{\text{pred}_t} = \text{LIU-Net}(X_t)$ |
| 25: | **end   for** |
| 26: | **Step 7:** Generate segmentation mask of the test set |
| 27: | Generate: Segmentation masks for the test set |
| 28: | **Step 8:** Test and generate the segmentation mask of the BraTS 2021 validation set |
| 29: | Evaluate: Test the model on the BraTS 2021 validation set |
| 30: | Generate: Segmentation masks for the BraTS 2021 validation set |
| 31: | **Step 9:** Cross-dataset validation on the BraTS 2020 training dataset |
| 32: | Read: BraTS 2020 training dataset |
| 33: | Evaluate: Test the model on the BraTS 2020 training dataset |
| 34: | **end** |

## Loss function

Numerous factors impact the performance of deep learning models, and selecting an appropriate loss function is a pivotal aspect that works hand in hand with the model's

architecture. The process of brain tumor segmentation poses a considerable challenge. Like the class imbalance issue presents a notable concern, the choice of an appropriate loss function holds great importance in tackling this issue and improving the overall performance and precision of the model in accurately delineating all tumor regions (*Raza et al., 2023*; *Shaikh et al., 2022*). This research utilizes the dice loss coefficient function during the training, which can be computed by Eq. (2).

$$\text{Dice Loss} = \frac{2 \sum_i (p_i \cdot g_i)}{\sum p_i^2 + \sum g_i^2} \tag{2}$$

where $p_i$ represents the predicted probability (or intensity) of a pixel being in the foreground called the predicted mask, and $g_i$ represents the actual ground truth probability (or intensity) of a pixel being in the foreground (in the ground truth mask). The summation operation is performed over all pixels in the image. In medical image segmentation, class imbalance is a significant challenge, particularly in brain tumor segmentation, where the tumor (foreground) occupies a much smaller region compared to the background. Standard loss functions like binary cross-entropy (BCE) often result in poor segmentation performance due to the dominance of background pixels. To address this issue, focal loss is employed, which modifies BCE to focus more on hard-to-classify pixels.

## Focal loss formulation

Focal loss (FL) is defined as:

$$FL(p_t) = -\alpha(1 - p_t)^\gamma \log(p_t) \tag{3}$$

where:

- $p_t$ is the predicted probability of the true class.
- $\alpha$ is the class balance parameter (typically 0.25).
- $\gamma$ is the focusing parameter, controlling the down-weighting of easy examples (commonly $\gamma = 2.0$).

For binary classification (*e.g.*, tumor *vs.* background), the focal loss can be expanded as:

$$FL(y, p) = -\alpha y(1 - p)^\gamma \log(p) - (1 - \alpha)(1 - y)p^\gamma \log(1 - p) \tag{4}$$

where:

- $y \in \{0, 1\}$ is the ground truth label.
- $p$ is the predicted probability of class 1.

For multi-class classification, the loss generalizes as:

$$FL(p_t) = -\alpha_t(1 - p_t)^\gamma \log(p_t) \tag{5}$$

where $\alpha_t$ is a class-wise weighting factor.

## Combining focal loss with dice loss

Since segmentation tasks often use the Dice score for evaluation, we can combine Focal Loss with Dice loss for improved performance:

$$\text{Total Loss} = \lambda_1 \times \text{Focal Loss} + \lambda_2 \times \text{Dice Loss} \tag{6}$$

where $\lambda_1$ and $\lambda_2$ are weighting factors.

Focal loss effectively handles class imbalance in medical image segmentation, particularly for brain tumor segmentation where tumor regions are small. By combining Focal loss with Dice loss, we can achieve better segmentation performance and robustness.

## RESULTS AND ANALYSIS

This section provides a thorough overview of the experimental setup and evaluation measures used to gauge the efficacy of the proposed LIU-Net model. It offers a comparative analysis with existing methods. Subsequently, it discusses the validation of the external cohort and computational complexity.

### Experimental setup

Python was employed to implement the proposed LIU-Net model. Training, validation, and testing were conducted using "Google Colab Pro" with a TESLA T4 GPU. In this context, the TensorFlow and Keras libraries were utilized, along with the Adam optimizer set to a learning rate of 0.0001. This configuration was combined with ReLU activation and batch normalization for this purpose. The model was trained on 100 epochs with a batch size of 2. The research conducted experiments utilizing the BraTS 2021 benchmark dataset. In this study, 70% of the dataset was allocated for training purposes, while 10% was reserved for validation and another 20% for testing. This approach was undertaken to ensure the integrity and reliability of the experimental results. Multiple experiments were conducted using the proposed technique to ascertain the most effective combination of hyperparameters. This investigation determined the most effective configuration for achieving the desired outcomes.

In the hyperparameter tuning process, the experiments utilized larger filters to capture extensive local information. Subsequently, a gradual reduction in filter width was implemented to narrow down the generated feature space and obtain more representative information. This approach was adopted to achieve a lightweight model. Furthermore, the optimization of the learning rate was performed through empirical tuning in various experiments to identify the most suitable learning rate value. The process began with a larger learning rate and iteratively reduced the learning rate value. This strategic adjustment facilitated a quicker convergence towards the global minima, enhancing the efficiency of the training process. The details about the hyper-parameters set during the proposed model training are given in Table 3. Hyperparameters were optimized for accuracy, stability, and computational efficiency through several trials. These selections ensure the LIU-Net architecture can process 3D MRI images and segment brain tumors well. This study chose the Adam optimizer because it can modify learning rates and employ computers quickly. It combines two famous optimizers' finest features. Adam is

**Table 3 Hyperparameters of the LIU-Net model.**

| Hyperparameter | Hyperparameter value |
|---|---|
| Input size | $128 \times 128 \times 128 \times 4$ |
| Batch size | 2 |
| Learning rate | 0.0001 |
| Activation function (Hidden Layers) | ReLU |
| Activation function (Output Layers) | Softmax |
| Optimizer | Adam |
| Loss function | Dice coefficient and focal loss |
| Epochs | 100 |
| Output size | $128 \times 128 \times 128 \times 4$ |

good at numerous tasks; thus, many deep-learning jobs utilize it by default. This study selected Adam over SGD due to its faster convergence and superior handling of sparse gradients. This study chose a batch size of two to update the model's variables more regularly, which can improve generalization. Larger batch sizes can speed up training, but they can hinder generalization. A smaller batch size helps to balance model training time and new data performance. This study selects 100 epochs to provide the model with sufficient training cycles to understand complex data trends. Experiments demonstrated that this quantity allowed the model to converge without over-fitting. This study selected rectified linear unit (ReLU) for the hidden layers since it fixes the disappearing gradient problem and speeds model convergence. The output layer used softmax to derive probability distributions over the classes, which is useful for jobs that need to split objects into multiple classes. The study chose Dice coefficient loss because it directly improves predicted segmentation-ground truth overlap. The Dice coefficient was used to evaluate the BraTS 2021 dataset to reduce class imbalance. In medical image segmentation, tumor regions are often smaller than healthy tissue (the majority class). Standard measurements such as accuracy may not be accurate in this situation. However, the Dice coefficient accounts for erroneous positives and negatives, giving a more accurate picture of how well the model functioned with the minority class.

## Evaluation measure

The assessment of the proposed methodology relies on Dice coefficient scores and accuracy metrics. The Dice coefficient score is the primary evaluation measure in brain tumor segmentation. It measures the extent of agreement between the predicted segmentation mask and the actual ground truth, considering both incorrect omissions and false inclusions. The Dice score ranges from 0 to 1, where higher values signify better segmentation results. In clinical scenarios, the proposed model's efficiency is measured by segmenting the regions of the tumor into three distinct sub-regions: the tumor core (TC), consisting of all tumor regions excluding edema; the whole tumor (WT), which encompasses all three cancerous regions; and the enhancing tumor (ET), which is visible in

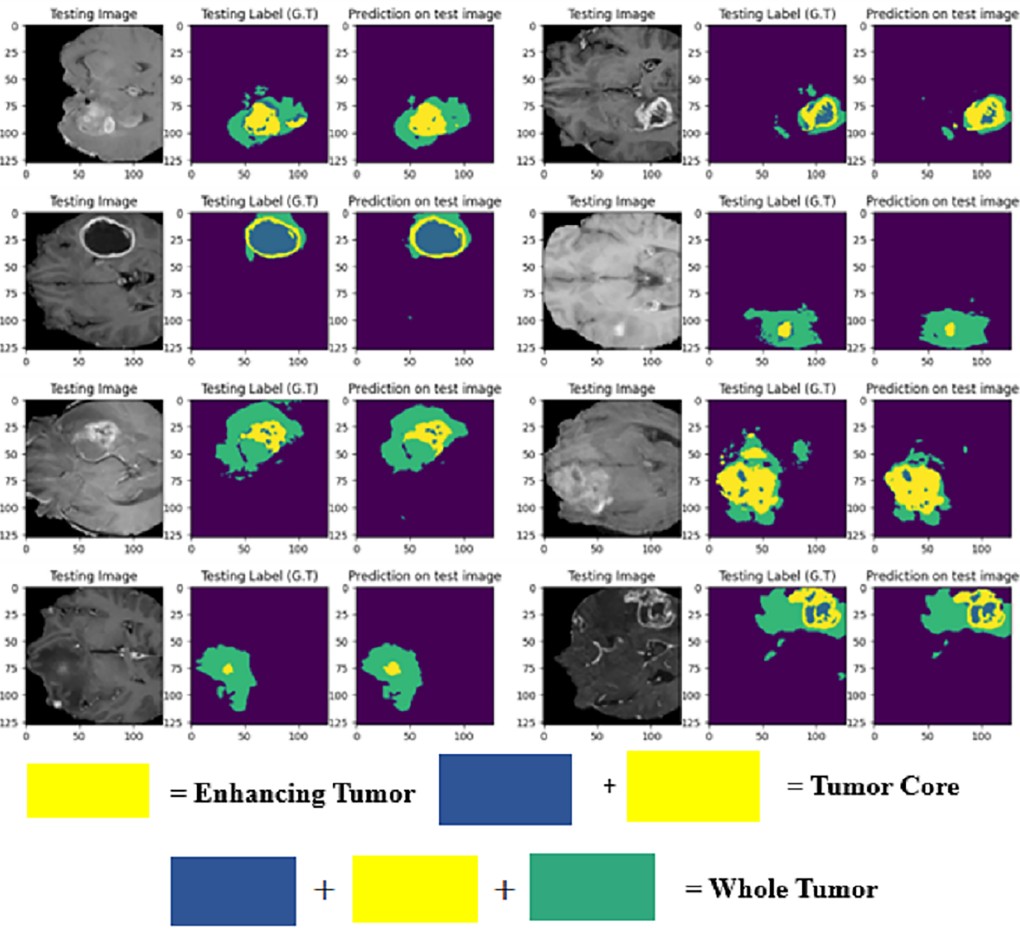

**Figure 6  Results of BTS on testing MRI sequences.**

the T1ce modality. In addition to the Dice coefficient score, accuracy metrics are employed to gauge the model's performance in accurately classifying and segmenting brain tumor sub-regions. The Dice score formula used in this research can be calculated with Eq. (7).

$$\text{Dice Score} = \frac{2|X \cup Y|}{|X| + |Y|} \tag{7}$$

## Results of the proposed model

The LIU-Net model was trained using the BraTS 2021 training dataset. The model underwent training on 70% of the dataset, validation on 10%, and was tested on the remaining 20% of the unseen data. The trained model was used to perform brain tumor segmentation on the test images. The described approach generates a three-dimensional (3D) representation that illustrates the segmentation mask covering the tumor areas known as WT, TC, and ET. For visualization, Fig. 6 displays the results of an MRI image observed in the axial plane, showcasing randomly selected slices. The visualization results declare the close alignment of the generated results with the ground truth for TC, WT, and

| Table 4 Results of the proposed architecture on the BraTS 2021 dataset. | | | |
|---|---|---|---|
| **BraTS 2021** | **WT** | **TC** | **ET** |
| Train set | 0.9498 | 0.9543 | 0.9033 |
| Validation set | 0.9027 | 0.9092 | 0.8646 |
| Test set | 0.8856 | 0.8444 | 0.8121 |

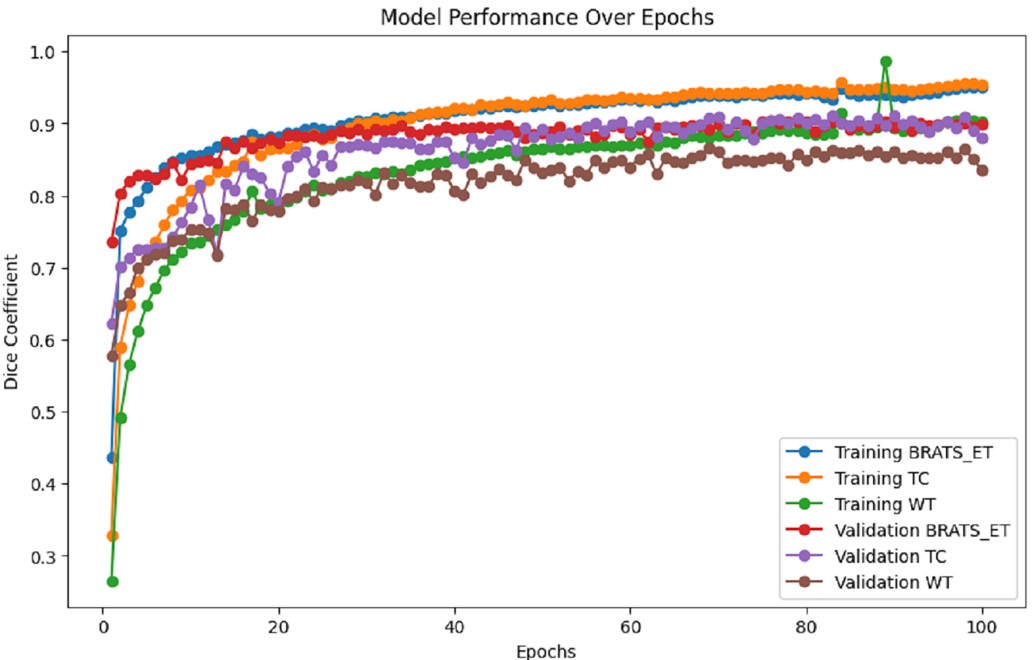

**Figure 7 Performance analysis of Dice coefficient.**

ET. This study displays LIU-Net's segmentation output on small and irregularly shaped tumors to demonstrate its performance in difficult conditions. These images show how LIU-Net can separate tumors accurately even when classes are imbalanced. Figure 6 illustrates the outcomes of an MRI image when observed in the axial plane with randomly selected slices. The yellow region represents the ET, the blue region corresponds to the non-enhancing TC, and the green region highlights the edema or peritumoral Region. The last colored block, labeled as the WT, is the union of these three subregions:

1. the non-enhancing TC (blue),
2. the ET (yellow), and
3. the edema (green).

Table 4 furnishes comprehensive details regarding the Dice coefficient values and accuracy metrics for ET, WT, and TC across the train set, validation set, and test set, as applied to the BraTS 2021 dataset.

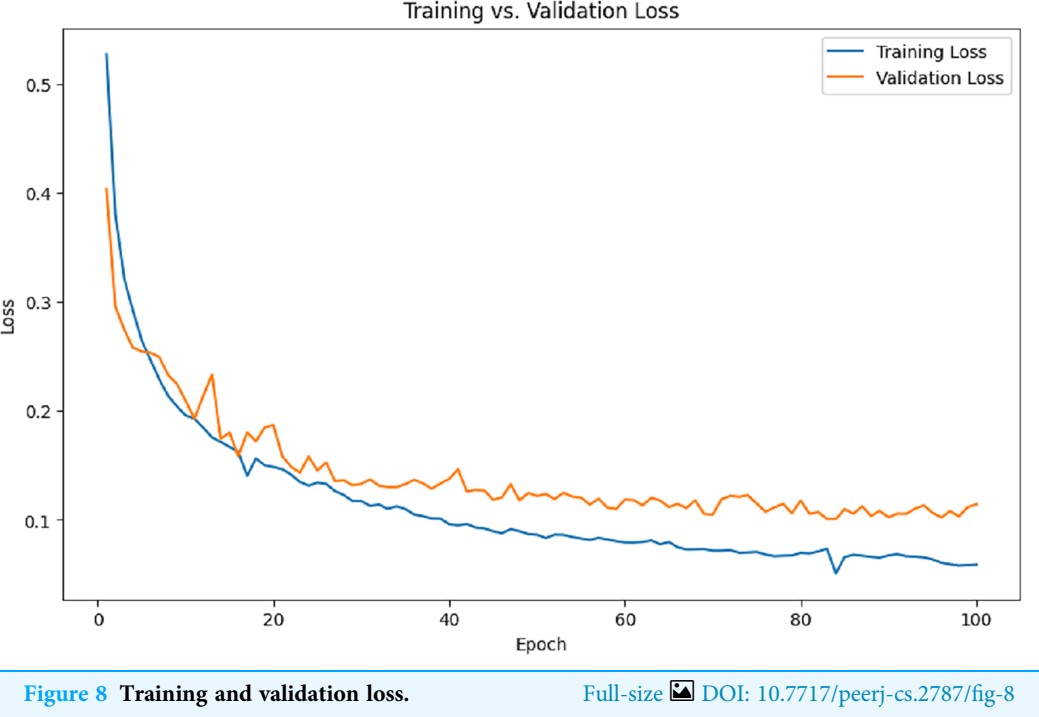

**Figure 8** **Training and validation loss.**

Figure 7 presents the training and validation metrics (WT, TC, and ET) plotted against the number of epochs for the proposed model. This visual representation illustrates the improvement in validation metrics over time in conjunction with the training metrics.

Figure 8 illustrates the training and validation loss per epoch. It is evident from the results that the training loss and validation loss steadily decrease over time.

This study represents progress in improving the efficiency of the U-Net model for the crucial task of brain tumor segmentation. While the existing U-Net model has historically demonstrated commendable performance, this study recognized the need to address certain challenges that could potentially hinder its utility, namely the issues related to over-parametrization and protracted training times. To tackle these challenges head-on, this research introduced a modified U-Net model with Inception blocks. This innovative approach, named LIU-Net, aims to optimize and streamline the segmentation process for brain tumors. By incorporating Inception blocks into the U-Net architecture, this study has strategically improved the model's capacity to capture intricate multiscale features and patterns within medical imaging data. The decision to integrate Inception blocks into the modified U-Net model stems from their proven efficacy in feature extraction and representation learning. This improvement is especially advantageous in medical imaging segmentation, given the subtle and intricate nature of tumor features requiring nuanced comprehension. The pursuit of this novel model was motivated by the desire to maintain the performance and effectiveness and elevate the quality of results achieved in brain tumor segmentation. By mitigating over-parametrization and reducing training time, the

**Table 5 Testing results on the BraTS 2021 augmented dataset using Dice loss.**

| BraTS 2021 | WT | TC | ET |
|---|---|---|---|
| Train set | 0.9389 | 0.9323 | 0.8933 |
| Validation set | 0.9134 | 0.9156 | 0.8876 |
| Test set | 0.8945 | 0.8500 | 0.8098 |

proposed modified LIU-Net promises to contribute to more efficient and accurate medical image analysis, ultimately benefiting patients and healthcare professionals.

## Results of the proposed model with data augmentation

The study further ran three tests to evaluate the model on data augmentation. Dice, Focal, and a mix of both were employed in these investigations. In the BraTS 2021 dataset, segmentation accuracy improved in all tumor areas: WT, TC, and ET.

### 1. Experiment with Dice loss

Initially, the research trained the LIU-Net model using Dice Loss as the main loss function and data improvement approaches. Using data enrichment improved Dice scores in training, validation, and test sets compared to the previous experiment. Table 5 furnishes comprehensive details regarding the Dice coefficient values and accuracy metrics for ET, WT, and TC across the train set, validation set, and test set, as applied to the BraTS 2021 dataset.

Data augmentation also provided enhanced model generalisation by reducing overfitting on the training set and boosting validation and test accuracy.

### 2. Experiment with Focal loss

Focal loss addresses uneven class distribution in classification tasks, notably with highly imbalanced data. This improves cross-entropy loss. It prioritises challenging cases above the correctly tagged examples. Mathematical description of focal loss is described in Eq. (4).

In the second experiment, the study used Focal loss to address the class imbalance in tumor segmentation. This approach assigns higher weights to misclassified pixels, ensuring better detection of small tumor regions. Table 6 furnishes comprehensive details regarding the Dice coefficient values and accuracy metrics for ET, WT, and TC across the train set, validation set, and test set, as applied to the BraTS 2021 dataset.

Focal Loss was better at identifying small, hard-to-find tumors than Dice loss. It performed slightly worse for larger tumor areas since it focused on problematic pixels.

### 3. Experiment with Combined Dice and Focal Loss

In the third experiment, the study combined Dice loss and Focal loss to leverage the strengths of both functions—Dice loss's ability to optimize segmentation boundaries and Focal loss's ability to handle class imbalance. Table 7 furnishes comprehensive details

**Table 6 Testing results on the BraTS 2021 augmented dataset using focal loss.**

| BraTS 2021 | WT | TC | ET |
|---|---|---|---|
| Train set | 0.9466 | 0.9455 | 0.9088 |
| Validation set | 0.9000 | 0.9200 | 0.8766 |
| Test set | 0.9087 | 0.8788 | 0.8233 |

**Table 7 Testing results on the BraTS 2021 augmented dataset using combined (Dice and Focal) loss.**

| BraTS 2021 | WT | TC | ET |
|---|---|---|---|
| Train set | 0.9099 | 0.9455 | 0.9233 |
| Validation set | 0.9677 | 0.9444 | 0.9067 |
| Test set | 0.9344 | 0.9555 | 0.9333 |

regarding the Dice coefficient values and accuracy metrics for ET, WT, and TC across the train set, validation set, and test set, as applied to the BraTS 2021 dataset.

The results show that this combined approach achieved the highest segmentation accuracy among all experiments. The model effectively handled class imbalances while maintaining precise tumor boundary delineation, making it the most robust and reliable configuration for brain tumor segmentation.

## Comparison with existing models

The LIU-Net model has been subjected to a comprehensive comparative analysis against existing models designed to segment brain tumors. In contrast to established segmentation techniques, the presented method demonstrates an improved capacity for accurately segmenting tumor regions that closely correspond to the ground truth, as substantiated by the research results. Table 8 presents a comparative analysis between the proposed LIU-Net model and existing models. The dataset column describes the dataset used in the comparative studies. Dataset 2020 and Dataset 2021 are the BraTS datasets used in the concerned studies.

The rationale behind selecting this architecture was to create a lightweight model that would efficiently utilize computational resources and ensure solving longer training time problems for comprehensive learning. Second, this research aimed to mitigate the issues associated with model over-parameterization, which can lead to increased computational demands and reduced generalization capability. The application of the LIU-Net produced promising outcomes, showcasing the effectiveness of the proposed method in achieving precise brain tumor segmentation. This decision not only highlights the dedication to enhancing computational efficiency but also underscores the commitment to generating significant outcomes in the realm of medical image analysis. This study examines WT, TC, and ET. The study compares the LIU-Net with several existing models. The comparison clearly shows the difference and effectiveness of the proposed model results from the existing ones. Axial Transformer (2021) had a WT score of 0.9321, while Redundancy

**Table 8 Comparative analysis.**

| References | Dataset | Model | WT | TC | ET |
|---|---|---|---|---|---|
| *Elmezain et al. (2022)* | BraTS 2021 | LDCRF | 0.8700 | 0.8500 | 0.8300 |
| *Peiris et al. (2022)* | BraTS 2021 | Adversarial | 0.9076 | 0.8500 | 0.8138 |
| *Wang & Dai (2023)* | BraTS 2021 | Swin UNETR | 0.9260 | 0.8850 | 0.8580 |
| *Liu et al. (2022)* | BraTS 2021 | SGEResU-Net | 0.9164 | 0.8685 | 0.8331 |
| *Raza et al. (2023)* | BraTS 2020 | dResU-Net | 0.8660 | 0.8357 | 0.8004 |
| *Jiang et al. (2022)* | BraTS 2020 | SwinBTS | 0.8906 | 0.8030 | 0.7736 |
| *Siddiquee & Myronenko (2021)* | BraTS 2021 | Redundancy | 0.9265 | 0.8868 | 0.8600 |
| *Wang et al. (2021)* | BraTS 2020 | TransBTS | 0.8900 | 0.8136 | 0.7850 |
| *Abd-Ellah et al. (2024)* | BraTS 2017 | TPCUAR-Net | 0.8700 | 0.8300 | 0.7600 |
| *Ren et al. (2024)* | BraTS 2023 | Transfer learning | 0.6764 | 0.7214 | 0.7159 |
| *Liu & Kiryu (2024)* | BraTS 2021 | Axial Transformer | 0.9321 | 0.9191 | 0.9006 |
| *Zhou et al. (2024)* | BraTS 2019 | MambaBTS | 0.8645 | 0.7350 | 0.8175 |
| *Luo et al. (2021)* | BraTS 2018 and 2017 | HDC-Net | 0.897 | 0.847 | 0.809 |
| *Liu & Kiryu (2024)* | BraTS 2018 | 3D Medical Axial Transformer | 0.9305 | 0.8791 | 0.8281 |
| *Chi et al. (2024)* | UCSF-PDGM, BraTS 2021 and 2019 | N-shaped lightweight | 0.9038 | 0.8749 | 0.8578 |
| **Proposed** | BraTS 2021 | LIU-Net | 0.9027 | 0.9092 | 0.8646 |

(2021) and Swin UNETR (2021) were close behind at 0.926. LIU-Net (2021) earned a WT score of 0.9027, placing it among the best models. Again, Axial Transformer (2021) had the highest TC score of 0.9191. In this category, LIU-Net (2021) scored well with 0.9092 TC. It accurately identified the TC as the second-best model for this metric. The highest score was 0.9006 for the Axial Transformer (2021), followed by Redundancy and Swin UNETR. LIU-Net (2021) has a 0.8646 ET score, ranking it among the best models. LIU-Net (2021) outperforms all except the axial transformer in TC measurements. In conclusion, LIU-Net's brain tumor separation results are strong and stable, making it a valuable field addition with fewer computational resources.

## External cohort validation using BraTS 2020 dataset

According to the insights presented in the studies (*Raza et al., 2023*), evaluating a model's performance on an unfamiliar dataset offers an unbiased assessment of its capabilities. While a model may demonstrate impressive performance on the training data, if it exhibits subpar results on an independent external dataset, it indicates overfitting to the training data. Overfit models have limited practical applicability in real-time scenarios. To validate the proposed LIU-Net model's robustness, it underwent external cohort validation using the BraTS 2020 dataset. The Dice coefficients obtained through external cohort validation for the proposed architecture yielded values of 0.8996, 0.8360, and 0.7925 for the WT, TC, and ET regions, respectively. A detailed breakdown of the external dataset experiment results is presented in Table 9.

The visualization provided breaks down LIU-Net's multiscale feature extraction. It shows how convolutional routes link Inception-style blocks. These pathways detect small,

**Table 9 External cohort validation results.**

| Method | Dataset | WT | TC | ET |
|---|---|---|---|---|
| External cohort validation | BraTS 2020 | 0.8360 | 0.8996 | 0.7925 |
| Test data | BraTS 2021 | 0.8444 | 0.8856 | 0.8121 |

medium, and large image features needed to identify cancers of various sizes. LIU-Net reliably handles tumor growth and shape variations by combining these properties. In medical imagery, where tumors can look, feel, and be quite different sizes, focusing on many scales is helpful. These results, along with Dice scores and IoU values, demonstrate that the approach may be utilized in clinical settings to segment difficult situations clearly and reliably. Figure 9 visually represents the MRI sequence results from the external cohort experiment, as observed in the axial plane. It showcases randomly selected slices from the BraTS 2020 dataset. The visual representation confirms that the obtained results closely align with the ground truth values for TC, ET, and WT.

As evident from the outcomes of external cohort validation, the proposed model demonstrates strong generalization capabilities when applied to previously unseen data. This substantiates the model's resilience and efficacy.

Validation was performed using 2021 test results and a 2020 external cohort. This study examines WT, TC, and ET. On the 2020 dataset, LIU-Net achieved a WT score of 0.8360 for external cohort validation. The WT score increased to 0.8444 on the 2021 test data, proving it works on all datasets. External cohort validation showed a high TC score of 0.8996, indicating that the model can isolate the TC. Despite a lower 2021 TC score of 0.8856, it performed well. ET = 0.7925 for outside group confirmation. The ET score rose to 0.8121 for the 2021 test data, indicating that the model can locate the ET region across datasets. The external cohort evaluation shows that the LIU-Net model reliably separates brain tumors across datasets. Because whole WT, TC, and ET performance measures are still good, the model functions well with new data. Scores above 0.83 indicate reliable and predictable performance in the WT dataset. The model's strong TC scores in both validations suggest it can identify the TC. Overall, external cohort validation findings confirm the LIU-Net model's reliability and stability. It can be beneficial for clinical brain tumor segmentation. The model was also utilized to generate predictions on the BraTS 2020 validation dataset, which lacks ground truth annotations. Figure 10 depicts the predictive results generated for the BraTS 2020 validation dataset, which lacked publicly available ground truth data. It can be confirmed from predicted results that the proposed LIU-Net model outperforms the validation dataset where ground truth masks are unavailable.

## Computational complexity

The proposed LIU-Net model has a total of 3.124 M parameters with a model size of 11.92 MB and approximately 58.66 Gega floating point operations per second (GFlops), which

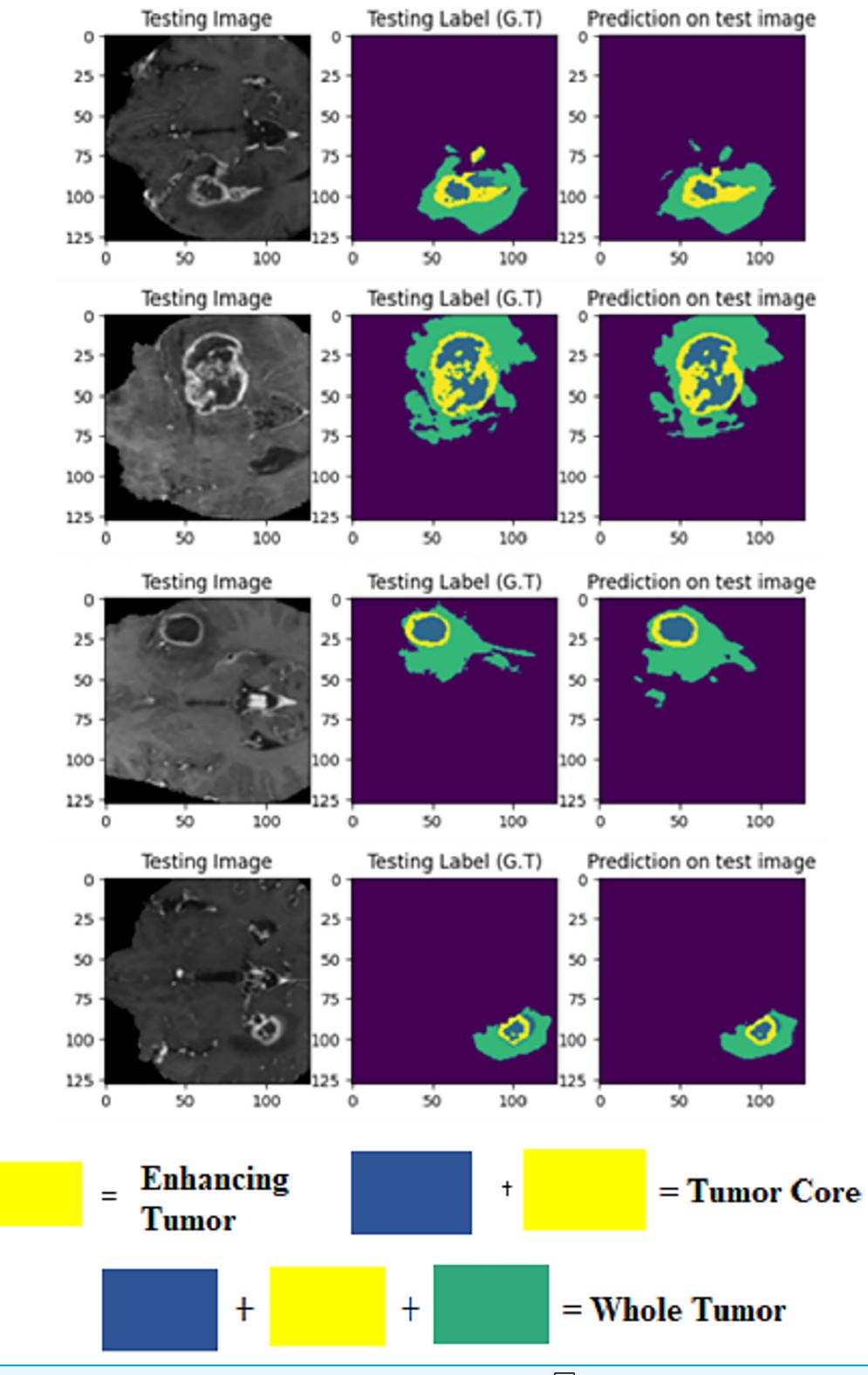

**Figure 9 Prediction results on BraTS 2020 dataset.**

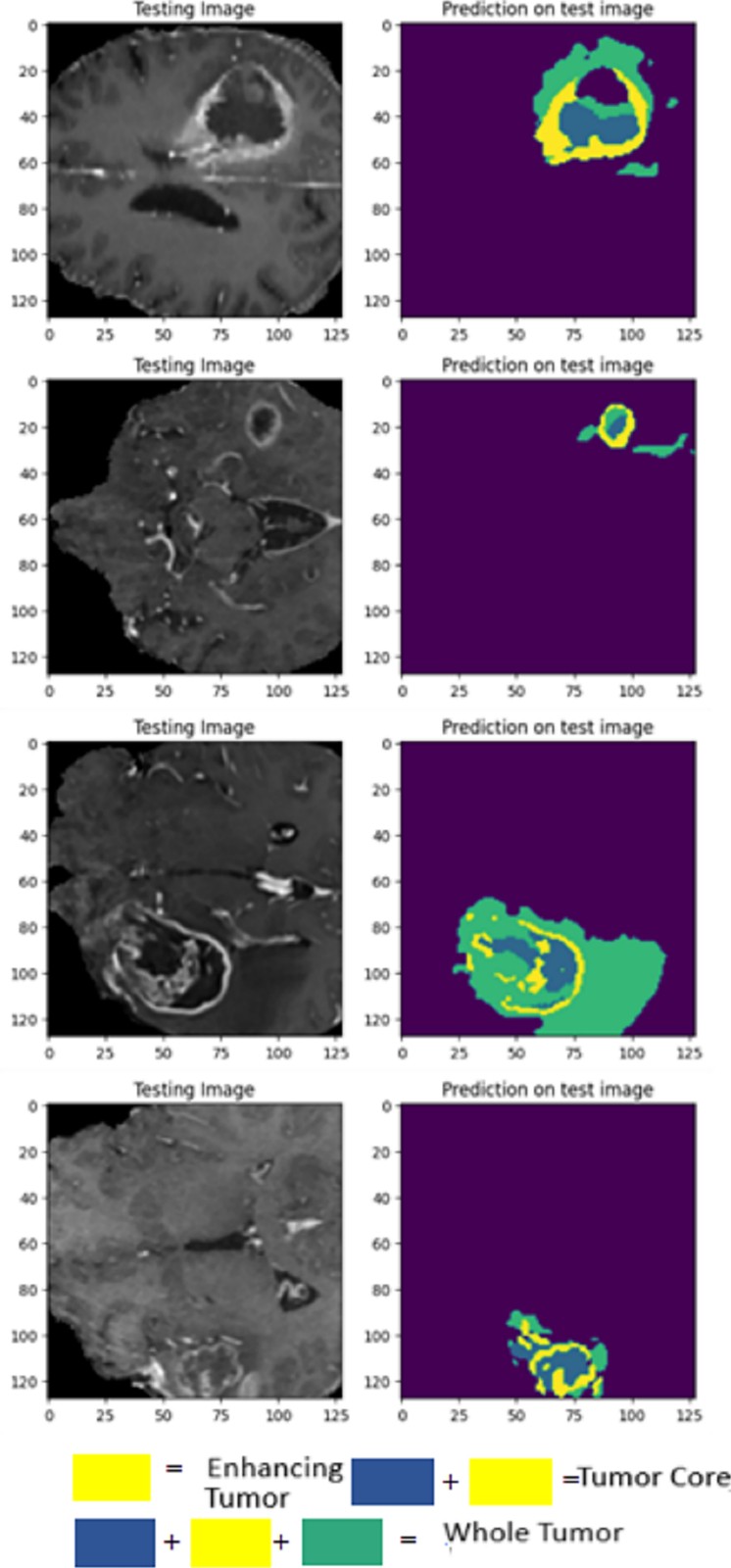

**Figure 10  Cross-validation on BraTS 2020 validation dataset.**

**Table 10 Summary of the computational resources.**

| Model | Dataset | Parameters (M) | FLOPs (G) | Training time | Testing time |
|---|---|---|---|---|---|
| U-Net | BraTS 2021 | 19.06 | 1670 | 58 h | 6 s |
| LIU-Net | BraTS 2021 | 3.124 | 58.66 | 20 h | 3 s |
| dResU-Net *Raza et al. (2023)* | BraTS 2020 | 30.47 | 374.04 | – | – |

makes the proposed model a lightweight model. The presented model is adaptable for application in a wide range of real-world clinical scenarios where computational efficiency is a priority. This efficient architecture allows for resource-conscious utilization in diverse contexts. The time required for executing the proposed model amounted to 15 min per epoch, and the training process encompassed 100 epochs, resulting in an overall training duration of approximately 20 h. The real-time assessment duration for the proposed model was remarkably swift, taking only 3 s for a single test subject, thus demonstrating its practical viability in clinical settings. Moreover, it is worth noting that the potential exists for further enhancing the model's performance by utilizing more robust computational resources. The summary of the computational resources of the proposed LIU-Net model and baseline U-Net model are provided in Table 10.

U-Net has 19.06 M parameters, and LIU-Net has 3.124. Fewer factors make LIU-Net lightweight and fast to compute. Multiply-Add Flops (MFlops) illustrate how difficult models are to compute. The U-Net has 1670 MFlops, while the LIU-Net has 58.66. This huge decline in MFlops demonstrates the computational power efficiency of LIU-Net. LIU-Net training lasts 20 h, whereas U-Net lasts 58. Low training time illustrates that LIU-Net uses duration and computer resources well, enabling faster development loops and iterations. LIU-Net tests samples in 3 s, whereas U-Net takes six. LIU-Net's faster reasoning time helps real-time applications draw speedy conclusions. The recommended LIU-Net model outperforms the U-Net model in computational complexity. With fewer parameters and computational power, LIU-Net learns and concludes faster. LIU-Net is ideal for clinical settings with few computers but high processing speeds. The TESLA T4 GPU, Intel Xeon CPU, and 32 GB of RAM provided enough computing capability to complete these tests swiftly.

## Ablation study

The LIU-Net achieved superior Dice scores compared to the baseline method for each specific brain tumor sub-region. The model obtained comparable results because of the incorporation of Inception blocks in the U-Net architecture. LIU-Net incorporates Inception-style blocks, featuring multiple convolutional blocks with varying kernel sizes ($1 \times 1 \times 1$, $3 \times 3 \times 3$, and $5 \times 5 \times 5$) to capture multi-scale hierarchical features effectively. The proposed model's design comprises an encoding path consisting of down-sampling layers and a decoding path equipped with up-sampling layers, enabling the model to acquire hierarchical representations of the input data. Each down-sampling block employs the Inception blocks, promoting extracting informative features at multiple scales.

**Table 11 Results of the ablation study on validation.**

| Model | WT | TC | ET |
|---|---|---|---|
| Baseline: 3D U-Net | 0.8531 | 0.8203 | 0.7581 |
| Proposed: LIU-Net | 0.9027 | 0.9092 | 0.8646 |

**Table 12 Results of paired sample t-test performed on BraTS 2021 dataset.**

| Brain tumor sub-region | T-statistic | *P*-value |
|---|---|---|
| ET | 5.45970 | 0.03195 |
| TC | 9.38040 | 0.01117 |
| WT | 7.08460 | 0.01935 |

Moreover, skip connections are established between the encoding and decoding path to facilitate the flow of detailed information during the up-sampling process.

Ablation experiments were carried out to assess the performance of the proposed LIU-Net model. The experiments highlight that Inception-style design and lightweight architecture are advantageous for certain medical imaging scenarios, particularly when dealing with complex anatomical structures. Furthermore, the proposed LIU-Net model exhibits superior computational efficiency compared to the baseline model. Training a single epoch of the LIU-Net takes approximately 15–16 min, unlike the baseline model, which requires 34–36 min to complete a single epoch. This significant reduction in training time underscores LIU-Net's practicality and effectiveness, making it an even more appealing choice for time-sensitive medical image segmentation tasks. Table 11 presents the impact of both methodologies on the Dice scores for the three sub-regions: ET, WT, and TC.

## Statistical analysis

This study further conducted paired sample t-tests to prove the statistical significance of the proposed model and to analyze the segmentation outcomes achieved by the LIU-Net model compared to the baseline U-Net model. The paired sample t-test was utilized for statistical analysis because it compares the means of similar groups or situations. This study compared the segmentation results of the suggested LIU-Net model and the traditional U-Net model. The analysis use the paired sample t-test to see if there are statistically significant differences in Dice scores between the ET, WT, and TC regions. For the same dataset, LIU-Net findings are directly compared to U-Net results. Since the recommended model and baseline model employ the same images, the paired sample t-test is useful. This enables the study to direct comparison of the two models' performance without considering dataset differences. This strategy considers the confusion that arises from the use of distinct data sources by both models. Statistics were based on 100 MRI scans. ET, WT, and TC segmentation results were collected using

LIU-Net and baseline U-Net models. The results, including the t-statistic and corresponding $p$-values, are summarized in Table 12. This table reveals that the $p$-values for the ET, WT, and TC regions were below the significance threshold of 0.05. Hence, based on our findings, we can confidently conclude that, at the 0.05 significance level, the segmentation Dice scores obtained with the proposed LIU-Net significantly surpassed those obtained with the baseline model. These outcomes underscore the efficacy of the proposed methodology, particularly in the context of real-time applications.

## CONCLUSION AND FUTURE DIRECTIONS

This work proposed an innovative LIU-Net architecture for BTS using multimodal 3D MRI images. Our investigation confronts the intricate complexities associated with model over-parameterization and prolonged training time, as observed in existing models. The conducted experiments provide strong evidence, highlighting the effectiveness of integrating Inception-style blocks into the U-Net architecture. This integration amplifies the model's efficiency across training duration and hyperparameter optimization, culminating in an enhanced tumor segmentation performance. The comprehensive evaluation on the BraTS 2021 dataset validates the potential of the LIU-Net model. The proposed model achieved a remarkable Dice scores of 0.8121 for the ET region, 0.8856 for the WT region, and 0.8444 for the TC region. These findings affirm the model's efficiency in delineating tumor boundaries and capturing the nuanced intricacies within tumor sub-regions. Moreover, external cohort validation was performed to establish the versatility and robustness of our proposed LIU-Net architecture, utilizing the BraTS 2020 dataset. Encompassing datasets from many medical centers for validation holds the promise of evaluating the methodology's universal applicability and effectiveness across divergent imaging protocols and patient cohorts. In the future, a promising avenue lies in the exploration of optimization techniques that facilitate real-time or near-real-time model inference. Such capabilities could be invaluable in clinical settings, expediting prompt decision-making processes.

### Funding

This article has been supported by Princess Nourah bint Abdulrahman University Researchers Supporting Project number (PNURSP2025R759), Princess Nourah bint Abdulrahman University, Riyadh, Saudi Arabia. The funders had no role in study design, data collection and analysis, decision to publish, or preparation of the manuscript.

### Grant Disclosures

The following grant information was disclosed by the authors:
Princess Nourah bint Abdulrahman University, Riyadh, Saudi Arabia: PNURSP2025R759.

### Competing Interests

The authors declare that they have no competing interests.

## Author Contributions

- Gul e Sehar Shahid conceived and designed the experiments, performed the experiments, analyzed the data, performed the computation work, prepared figures and/or tables, authored or reviewed drafts of the article, and approved the final draft.
- Jameel Ahmad conceived and designed the experiments, analyzed the data, performed the computation work, prepared figures and/or tables, authored or reviewed drafts of the article, and approved the final draft.
- Chaudary Atif Raza Warraich conceived and designed the experiments, analyzed the data, performed the computation work, prepared figures and/or tables, authored or reviewed drafts of the article, and approved the final draft.
- Amel Ksibi conceived and designed the experiments, performed the experiments, analyzed the data, prepared figures and/or tables, authored or reviewed drafts of the article, and approved the final draft.
- Shrooq Alsenan performed the experiments, analyzed the data, performed the computation work, authored or reviewed drafts of the article, and approved the final draft.
- Arfan Arshad performed the experiments, analyzed the data, performed the computation work, authored or reviewed drafts of the article, and approved the final draft.
- Rehan Raza analyzed the data, performed the computation work, prepared figures and/or tables, authored or reviewed drafts of the article, and approved the final draft.
- Zaffar Ahmed Shaikh analyzed the data, performed the computation work, prepared figures and/or tables, authored or reviewed drafts of the article, and approved the final draft.

## Data Availability

The benchmark BraTS dataset is available at https://www.med.upenn.edu/cbica/brats.

For access to the proposed framework and additional technical details of this research, please refer to https://github.com/rehanrazaa/LIU-Net-Lightweight_Inception_U-Net_Brain_Tumor_Segmentation_BraTS2021.

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
