# Peer review of "LIU-NET: lightweight Inception U-Net for efficient brain tumor segmentation from multimodal 3D MRI images"

_PeerJ Computer Science, doi:10.7717/peerj-cs.2787_

## Round 0.1 · original submission · Major Revisions

Dear authors,

You are advised to critically respond to all comments point by point when preparing an updated version of the manuscript and while preparing for the rebuttal letter. Please address all comments/suggestions provided by reviewers, considering that these should be added to the new version of the manuscript.

Some reviewers suggested that you cite specific references. You are welcome to add it/them if you believe they are relevant. However, you are not required to include these citations, and if you do not include them, this will not influence my decision.

Kind regards,
PCoelho

Reviewer 1 ·

Basic reporting

The paper is generally well-written, but some sections could benefit from clearer language, particularly in explaining technical terms. For example, "balanced stability" in relation to model complexity and computational load could be phrased more clearly to enhance readability.
Consistency in terminology should be maintained, especially in describing model components (e.g., "Inception-style blocks" vs. "Inception modules").

The introduction provides a solid foundation on the challenges in brain tumor segmentation and the limitations of existing models, but it would be beneficial to include a brief summary of how LIU-Net addresses each challenge, particularly computational complexity, multiscale feature extraction, and class imbalance.
Citing a few more recent and relevant studies on lightweight models in medical imaging would help readers understand where LIU-Net stands within current research.

The literature review is adequate but could be enhanced by discussing more recent lightweight models or architectures specifically used for medical image segmentation. This would provide readers with a better understanding of how LIU-Net compares to similar work.
The paper would benefit from clarifying the novelty of the Inception blocks in the U-Net architecture, as well as any specific advantages they bring in comparison to previous work.

Experimental design

Strengths
The experiments conducted on BraTS 2021 with cross-validation on BraTS 2020 provide a solid foundation, as these datasets are widely used and highly regarded in the medical imaging field. This choice strengthens the validity of the results and allows for fair comparisons with other studies.

The inclusion of Dice scores across different regions — Enhancing Tumor (ET), Whole Tumor (WT), and Tumor Core (TC) — is well-suited for this application, as each region has unique segmentation challenges. This choice allows for a nuanced understanding of LIU-Net’s effectiveness on different tumor areas.

The authors not only assess segmentation accuracy but also measure computational efficiency, such as parameter count and training time, which is essential for a lightweight model designed for clinical applications.

Incorporating a weighted Dice coefficient as part of the loss function addresses class imbalance, a common issue in medical segmentation tasks. This approach is thoughtful and improves the model’s ability to handle underrepresented classes effectively.

Areas for Improvement
The experiments could be enhanced by comparing LIU-Net not only with heavy models but also with other lightweight or efficiency-focused architectures. This would provide more insight into where LIU-Net stands among models with similar goals of balancing accuracy and computational cost.
The paper could strengthen its claims by clarifying the cross-validation approach on BraTS 2021 (e.g., K-fold or leave-one-out validation). Additionally, if computational resources allow, repeating the experiments across multiple splits could improve the reliability of the reported results.

Validity of the findings

The findings presented in this paper are promising and indicate that LIU-Net is a viable model for brain tumor segmentation. However, there are some considerations that impact the robustness and generalizability of the conclusions.

Positive Aspects
The model’s performance on the BraTS 2021 and BraTS 2020 datasets lends credibility to the reported results, as these are widely accepted benchmarks for brain tumor segmentation. This helps in validating LIU-Net’s effectiveness against well-known datasets with diverse, real-world examples of brain tumor imagery.

The Dice score is an appropriate and widely accepted metric for evaluating segmentation accuracy in medical imaging. Reporting Dice scores for different tumor regions (Enhancing Tumor, Whole Tumor, Tumor Core) provides a detailed view of the model’s performance, making the findings more transparent and meaningful.

The use of cross-validation on an external cohort (BraTS 2020) supports the findings' validity, as it shows that LIU-Net’s performance is not limited to a single dataset. This strengthens the claim of model generalizability and robustness.

Areas for Further Validation
Although the findings show that LIU-Net achieves high segmentation accuracy with fewer parameters, comparisons with additional, lightweight or efficient segmentation models would provide a stronger basis for the claim that LIU-Net is both accurate and computationally efficient. Without these comparisons, it is challenging to determine if LIU-Net truly offers a significant improvement over existing approaches in terms of the accuracy-complexity trade-off.

While the BraTS datasets are comprehensive, testing LIU-Net on additional MRI datasets with different imaging conditions or populations could further validate its robustness and adaptability to a wider range of clinical scenarios. Brain MRI data can vary significantly, and additional tests would make the findings more broadly applicable.

: The paper mentions a weighted Dice coefficient to address class imbalance but does not explore this component’s specific impact on model performance. Additional analysis or ablation studies showing how much this loss function improves segmentation performance, particularly for smaller or challenging tumor regions, would add credibility to this aspect of the findings.

Interpre Including visualizations of LIU-Net’s segmentation output on challenging cases (e.g., small or irregularly shaped tumors) and a breakdown of its multiscale feature extraction process would provide insights into how the model achieves its high Dice scores. This would also make the findings more transparent and interpretable for clinical audiences

Additional comments

NIIL

Annotated reviews are not available for download in order to protect the identity of reviewers who chose to remain anonymous.

Reviewer 2 ·

Basic reporting

Title: LIU-NET: Lightweight inception U-Net for eûcient brain tumor segmentation from multimodal 3D MRI images

1. Novelty need to be highlight with abstract
2. I could not find any difference between motivation and innovation chapters. Innovation should highlights the article novelty
3. The qualitative results of pre-processing (normalization and resizing) is required
4. BraTS having fixed size images. Then why u need resize the images? Is resize affects the diagnosis process?
5. If possible add more evaluation metrics, like AUC curve etc…
6. Fig.6, 9 and 10 quality is very poor
7. Results and discussion should be qualitative followed by the quantitative results.
8. Why t test is chosen for Statistical Analysis? What is the sample size?

Experimental design

Good

Validity of the findings

Good

Additional comments

Nil

Reviewer 3 ·

Basic reporting

1. Explain why the current method was selected for the study, and its importance and compare it with traditional methods.
2. The color description in Figs 6 and 9 is wrong.
3. In the literature, each paper should specify the proposed methodology, novelty, and results.
4. The paper lacks a summary of the method before presenting the details. This makes it complex to read and understand
5. The author should compare the proposed algorithm with other recent works or provide a discussion. Otherwise, it's hard for the reader to identify the novelty and contribution of this work
6. Fig. 1 should be presented in a more scientific style. Now it looks good for a scientific-popular magazine but not for a research journal.
7. Please imply the effect of the loss function on your study.

Experimental design

As above

Validity of the findings

As above

Additional comments

As above

Reviewer 4 ·

Basic reporting

I cannot recommend this paper for publication as you propose something that many other others have already done. To use Inception with U-Net has been done many times, and you do not cite a single of all papers doing that. Before you start a research project you need to search for papers on a similar topic.

Zhang, Z., Wu, C., Coleman, S., & Kerr, D. (2020). DENSE-INception U-net for medical image segmentation. Computer methods and programs in biomedicine, 192, 105395.

Punn, N. S., & Agarwal, S. (2020). Inception u-net architecture for semantic segmentation to identify nuclei in microscopy cell images. ACM Transactions on Multimedia Computing, Communications, and Applications (TOMM), 16(1), 1-15.

Sariturk, B., & Seker, D. Z. (2022). A residual-inception U-Net (RIU-Net) approach and comparisons with U-shaped CNN and transformer models for building segmentation from high-resolution satellite images. Sensors, 22(19), 7624.

Aboussaleh, I., Riffi, J., Mahraz, A. M., & Tairi, H. (2024). Inception-UDet: an improved U-Net architecture for brain tumor segmentation. Annals of Data Science, 11(3), 831-853.

Cahall, D. E., Rasool, G., Bouaynaya, N. C., & Fathallah-Shaykh, H. M. (2021). Dilated inception U-net (DIU-net) for brain tumor segmentation. arXiv preprint arXiv:2108.06772.

Banerjee, S., Lyu, J., Huang, Z., Leung, F. H., Lee, T., Yang, D., ... & Ling, S. H. (2022). Ultrasound spine image segmentation using multi-scale feature fusion skip-inception U-Net (SIU-Net). Biocybernetics and Biomedical Engineering, 42(1), 341-361.

Zong, Y., Chen, J., Yang, L., Tao, S., Aoma, C., Zhao, J., & Wang, S. (2020). U-net based method for automatic hard exudates segmentation in fundus images using inception module and residual connection. IEEE Access, 8, 167225-167235.

Wang, Y., Qin, C., Lin, C., Lin, D., Xu, M., Luo, X., ... & Ni, D. (2020). 3D Inception U‐net with asymmetric loss for cancer detection in automated breast ultrasound. Medical Physics, 47(11), 5582-5591.


Siciarz, P., & McCurdy, B. (2022). U-net architecture with embedded Inception-ResNet-v2 image encoding modules for automatic segmentation of organs-at-risk in head and neck cancer radiation therapy based on computed tomography scans. Physics in Medicine & Biology, 67(11), 115007.
* * *
"The U-Net architecture was originally introduced by Ronneberger et al. Aboelenein et al. (2020) and
gained widespread recognition for its capacity to produce reliable segmentation outcomes, particularly in scenarios with limited training data."

The U-Net paper was published in 2015 and not in 2020, why are you mentioning Aboelenein here ?
* * *
Since you are not using vision transformers, it seems strange to have such a long section about vision transformers.

Experimental design

Since you claim that your network is light weight, you should also compare the number of trainable parameters of ALL other models that you list Dice scores for, and the training time and inference time of all models. How did you decide that your U-Net for comparison should have 19 million parameters?

You do not mention if augmentation was used, it is very common to do image augmentation during training to further increase the training set.

You write that you use a weighted segmentation loss, but I don't see any weights for each segmentation class. Normally you need to put a much lower weight on the background for example.

Validity of the findings

You cannot directly compare different methods if you don't use exactly the same training and test set. If you want a fair comparison, you need to run all other models using your exact split of the data. Or use the official BraTS page where you can upload segmentations for getting Dice scores. Did you run all models yourself, or did you just take the test results from their papers? Do you even know if the other papers had the same size of their test set?

There is a large overlap between BraTS 2020 and 2021, so it does not make sense to use BraTS 2020 for validation.

Models should also be compared using the Hausdorff distance, and not only using Dice.

Did you train a single time or did you perform any cross validation?

Reviewer 5 ·

Basic reporting

The following points need to be addressed in the revision
Abstract
1. No novice idea was found in the work, although many claims were made.
2. The initial part needs to be trimmed, carrying introductory text.
3. The motivation of the study needs more clarity.
Introduction
1. The main innovation and contribution of this research should be clarified in the introduction.
2. Much of the introduction is irrelevant. Please try to present a review of the articles related to the segmentation/classification of brain tumors.
3. The articles are less relevant to the topic of interest. Improve the introduction by adding recent articles related to brain tumor segmentation.


1. Section 1.3: Contribution number 2 is a part of contribution number 1. The third contribution is part of the analysis. Do not consider it a contribution. Add more relevant contributions that are stand-alone.
2. The authors could better explain how "Related works" is related to the current study. It is unclear to the reader how the manuscript is similar to or differs from these related works.
Materials and Methods
1. Figure 1 needs improvement in multiple places. How is the loss function used after reaching a dead end? No testing model exists. The trained model from training is used for testing. How were the results validated? Improve the figure and add it to the text. Use 10-fold validation.
2. The Brats2021 needs more details to be added to the text portion.
3. Check the dimensional stability of the architecture in Figure 4.
Results and Analysis
1. Table 2 needs to be explained in the article. Explain input and outsizes. Explain why the output is not in a mask state.
2. Table 3 shows weak results. The article explains this point by comparing it with other work and standard low values of DC in this dataset.
3. Figure 6 needs an explanation for the last blocks. The MRI scans pasted in this figure need to be improved in quality.
4. Explain the formation of Table 4 for the source of the results. What validation and preprocessing were adopted?
5. Section 4.5: Wrong order used while representing results. The training source is not clear. Why comes the validation here? You were supposed to be connected with testing using a dataset. Explain the Table 5 thoroughly.
6. How cross-validation was carried out?
References
1. There is some problem with the references. Correct it.

Experimental design

As above

Validity of the findings

As above

Additional comments

The comments are mixed up. Any one of the lists can be followed to improve the article, keeping hybrid improvements in view.

---

## Round 0.2 · Minor Revisions

Dear authors,

Thanks a lot for your efforts to improve the manuscript.

Nevertheless, some concerns from Reviewer 4 are still remaining that need to be addressed regarding the data imbalance.

Like before, you are advised to critically respond to the remaining comments point by point when preparing a new version of the manuscript and while preparing for the rebuttal letter.

Kind regards,
PCoelho

Reviewer 1 ·

Basic reporting

Clarity and Structure: The revised paper is well-organized, and the language is clear and professional. The problem statement, methodology, and results are effectively articulated.
Figures and Visualizations: The inclusion of segmentation performance metrics is commendable

Experimental design

Appropriateness of the Approach: The experimental design is solid, with a well-defined methodology incorporating multiscale feature extraction using Inception blocks and addressing class imbalance through a weighted Dice loss function.
Dataset Selection: The use of BraTS 2021 and external validation on BraTS 2020 datasets is appropriate and helps validate the model's effectiveness.

Validity of the findings

Robustness of Results: The model's performance on BraTS datasets, with detailed segmentation metrics for enhancing tumor (ET), whole tumor (WT), and tumor core (TC), is convincing.
Addressing Research Gaps: The authors adequately address computational efficiency and multiscale feature extraction

Reviewer 2 ·

Basic reporting

-

Experimental design

-

Validity of the findings

-

Additional comments

-

Reviewer 3 ·

Basic reporting

According to the response letter, the paper has been revised, and the current version of the manuscript is acceptable for publication.

Experimental design

According to the response letter, the paper has been revised, and the current version of the manuscript is acceptable for publication.

Validity of the findings

According to the response letter, the paper has been revised, and the current version of the manuscript is acceptable for publication.

Reviewer 4 ·

Basic reporting

I still cannot recommend this paper to be accepted, because there is nothing new in this paper compared to previously published papers on the same topic. Furthermore the authors have ignored most of my requested changes.

You should still use augmentation to further improve the model.

No you cannot use un-weighted Dice for tumor segmentation, it does not inherently adress class imbalance.

Experimental design

See above

Validity of the findings

See above

Reviewer 5 ·

Basic reporting

The article may be accepted.

Experimental design

It was ok.

Validity of the findings

Findings seem valid.

Additional comments

Nil

---

## Round 0.3 · Major Revisions

Dear authors,

After the previous revision round, some adjustments still need to be made. As a result, I once more suggest that you thoroughly follow the instructions provided by the reviewer to answer their inquiries clearly.

Kind regards,
PCoelho

Reviewer 4 ·

Basic reporting

I still don't see how this paper is different to other papers which used Inception together with U-Net. You need to explain what is unique with your paper, after two revisions I still don't see what is unique here.

What is the difference between your work and references 9, 32, 41, 44 ? You need to write

"Compared to other works which also combined U-Net with inception (9,32, 41, 44), the difference in this work is bla bla bla "

I will not review this paper again, as you don't care about my comments.

Experimental design

.

Validity of the findings

.

Additional comments

.

---

## Round 0.4 · accepted · Accept

Dear authors, we are pleased to verify that you meet the reviewer's valuable feedback to improve your research.

Thank you for considering PeerJ Computer Science and submitting your work.

Kind regards
PCoelho